# Reduced utilitarian willingness to violate personal rights during the COVID-19 pandemic

Rea Antoniou[1]*, Heather Romero-Kornblum[1,2], J. Clayton Young[1], Michelle You[1,3], Joel H. Kramer[1], Winston Chiong[1]

**1** Department of Neurology, Memory and Aging Center, University of California, San Francisco, San Francisco, California, United States of America, **2** Rady School of Management, University of California, San Diego, San Diego, California, United States of America, **3** School of Medicine, New York Medical College, Valhalla, NY, United States of America

* Rea.Antoniou@ucsf.edu

**Data Availability Statement:** De-identified behavioral data (https://osf.io/w9r4b/) are available in the Open Science Framework repository.

## Abstract

The COVID-19 pandemic poses many real-world moral dilemmas, which can pit the needs and rights of the many against the needs and rights of the few. We investigated moral judgments in the context of the contemporary global crisis among older adults, who are at greatest personal risk from the pandemic. We hypothesized that during this pandemic, individuals would give fewer utilitarian responses to hypothetical dilemmas, accompanied by higher levels of confidence and emotion elicitation. Our pre-registered analysis (https://osf.io/g2wtp) involved two waves of data collection, before (2014) and during (2020) the COVID-19 pandemic, regarding three categories of moral dilemmas (personal rights, agent-centered permissions, and special obligations). While utilitarian responses considered across all categories of dilemma did not differ, participants during the 2020 wave gave fewer utilitarian responses to dilemmas involving personal rights; that is, they were less willing to violate the personal rights of others to produce the best overall outcomes.

## Introduction

At the time of this writing, there have been over 222 million global confirmed cases of COVID-19 (i.e., Coronavirus Disease 2019 caused by the novel coronavirus SARS-CoV-2) and over 4 million deaths [1]. The risk for severe illness from COVID-19 increases with age, with older adults being at highest risk for death and major disability [2]. As health systems and economies have been overwhelmed in many countries [3], the pandemic has presented policy-makers, medical professionals, and laypeople with stark moral dilemmas that often pit the needs and rights of the many against the needs and rights of the few. Understanding how individuals' moral reasoning is affected by global crises, such as the COVID-19 pandemic, has implications for public policy and communication about health measures.

In addition, many psychological and philosophical theories regarding human moral decision-making offer predictions about how individuals' moral reasoning is affected by real-

**Funding:** The authors received funding by the National Institutes of Health-National Institute on Aging (R01AG058817 and R01AG022983) and by the Larry L. Hillblom Foundation (2018-A-006-NET).The funders had no role in study design, data collection and analysis, decision to publish, or preparation of the manuscript.

**Competing interests:** The authors have declared that no competing interests exist.

world crises. In a recent article advocating for utilitarian approaches to this crisis, Savulescu and colleagues [4] have claimed that "There are no egalitarians in a pandemic," suggesting the empirical psychological claim that health crises may reduce the salience of non-utilitarian moral considerations. However, some influential accounts of moral cognition suggest instead that broad psychological effects in such a crisis [5–7], such as perceived threat or mortality salience, would lead people to adopt fewer utilitarian approaches to moral problems. Investigating changes in moral intuitions during the current pandemic, particularly among those (such as older adults) who are at greatest personal risk, can allow us to test the predictions of these theories.

## Differentiating utilitarianism from common-sense morality

In everyday life, we often appeal to what Parfit [8] calls "common-sense morality"—a morality defined by our relationships to particular people, such as our children, parents, friends, patients or fellow-citizens. Most societies forbid actions such as lying, cheating, stealing, injuring, and killing directed against community members. In contrast with common-sense morality, utilitarianism judges actions solely according to whether they produce the best outcomes, considered impartially.

Utilitarianism departs from common-sense morality in at least three respects [9]. First, in utilitarianism, pursuit of the best overall outcomes is not constrained by respect for *personal rights*. In situations where the best outcome can only be produced by violating another individual's rights (such as by harming or killing one person to save five other innocent people), utilitarianism treats such actions as permissible or even obligatory. Second, utilitarianism is highly "demanding," requiring agents to give their own interests no greater weight than other people; thus, often requiring great altruism (e.g., to donate almost all of one's material possessions to charity). In contrast, common-sense morality incorporates *agent-centered permissions* allowing agents to give greater weight to their own personal interests. Finally, the same spirit of impartiality means that utilitarianism does not encompass *special obligations*, which in common-sense morality direct agents to give greater weight to the interests of their own family, friends, patients, clients, students or compatriots than to the interests of those to whom they lack such ties.

## Utilitarian and non-utilitarian judgments

Most empirical moral psychological research has focused on contrasting utilitarian and non-utilitarian responses to hypothetical dilemmas involving personal rights [10], also referred to as "personal moral dilemmas" [11] or "sacrificial moral dilemmas" [12]. For example: *An epidemic has spread worldwide, killing millions of people. You have developed two substances in your underground shelter. One of them is a cure, but the other one is deadly. You do not know which is which. Two people have run downstairs to your shelter, trying to avoid the epidemic. The only way to identify the cure is to inject each of these people with one of the two substances. One person will live, but the other will die. Then you will be able to start saving lives with the cure. Would you kill one of these people with a deadly injection to identify a cure that will save millions of lives*?

While such scenarios have been critiqued as "complex, far-fetched and often convoluted," [12] they have proven highly influential in part because they have robust and reliable neural and psychological effects, in many cases yielding surprising insights into fundamental aspects of neural organization and mental processes [10, 13]. But while some individuals and clinical populations give responses to such dilemmas that are more consistent with utilitarianism than others', it remains controversial whether such responses represent general utilitarian patterns

of moral reasoning. Some evidence suggests that willingness to violate personal rights may be dissociable from impartial concern (as present in utilitarian judgments about dilemmas involving agent-centered permissions and special obligations) [14].

More broadly, making hypothetical decisions regarding particular cases may not entail that a decision-maker accepts the broader tenets of any ethical theory [12]. For the purposes of this and other empirical studies involving hypothetical cases, calling a decision "utilitarian" may only mean that this decision is consistent with utilitarianism, and may leave open various motivational or psychological outlooks that could underlie such choices.

## Dual-process theories in moral cognition

One influential account of individual differences in moral reasoning is Greene's dual-process theory [11, 13]. Informed by broader dual-process models of "intuitive" fast judgments and "deliberative" slow judgments [15, 16], on Greene's account characteristically deontological (non-utilitarian) judgments are driven by automatic emotional responses, while characteristically utilitarian judgments are driven by controlled cognitive processes. According to Greene's dual-process model of moral cognition, decisions to avoid causing harm, such as deciding not to administer a lethal injection, reflect more emotionally laden responses to the harm in question; whereas decisions that accept harm to minimize net suffering (thus optimizing aggregate welfare), such as lessening the impact of the epidemic, would reflect more deliberative cognitive control processes [17].

Decisions in personal rights dilemmas are associated with other psychological and emotional phenomena in decision-makers. Prior work [18] suggests that post-decisional emotions elicited by utilitarian options are stronger in moral dilemmas that involve delivering harm personally as opposed to impersonally. In addition, emotions associated with counterfactual comparisons drive decision-making in personal rights moral dilemmas, as people find it aversive both to cause harm [19] and to witness others' suffering [20]. The ensuing emotions can reduce willingness to cause harm [21], while emotion reappraisal or suppression can attenuate the impact of emotion on non-utilitarian (e.g., harm-avoidance) judgments [16, 21, 22].

## Terror management theory and threat perception

The current threat of the COVID-19 outbreak is likely to enhance the emotional salience of death and other personal risks. Prior research drawing on terror management theory suggests that individuals who are primed with thoughts of impending death are less likely to give utilitarian responses on moral conflicts [6]. In a similar vein, disease threat perception is related to sensitivity towards moral violations [5, 7] in domains consistent with common-sense morality (i.e., binding moral domains).

Terror management theory additionally posits that death-related thoughts may lead to existential fear, which can be attenuated by maintaining faith in an internalized cultural worldview. Accordingly, seeking emotional safety in one's worldview leads to greater confidence in one's values [23]. Older adults, who are at higher risk for death and disability from COVID-19, may be especially likely to demonstrate these effects.

## Hypotheses and research questions

The current study utilizes a unique opportunity to study utilitarian judgments within the context of COVID-19 pandemic. In 2014, we collected older adults' responses to a series of hypothetical moral dilemmas as part of piloting an online testing platform in an existing research cohort. In March 2020, we preregistered (https://osf.io/g2wtp) a new study utilizing these data by delivering the same moral dilemma instrument to older adults during the 2020 COVID-19

pandemic. This population is of particular interest because older adults are at greatest risk of death and major disability due to this infection [2].

We hypothesized that individuals would give fewer utilitarian responses to moral dilemmas during the 2020 COVID-19 pandemic than prior to this outbreak (Hypothesis 1) and would rate their decisions as made with greater confidence (Hypothesis 2) consistent with terror management theories. Last, we hypothesized that ratings of emotion elicitation would be higher (Hypothesis 3), consistent with dual-process theories.

In our pre-registered primary analyses (https://osf.io/g2wtp), we considered together utilitarian responses to three categories of hypothetical dilemmas: those involving personal rights, agent-centered permissions, and special obligations. Because dilemmas involving personal rights have been the primary focus of the literature to date, and given controversies over the degree to which utilitarian responses to these dilemmas rely upon the same psychological mechanisms as utilitarian responses to dilemmas involving agent-centered permissions and special obligations [14], we also pre-registered planned subsidiary analyses focusing exclusively on the subset of dilemmas involving personal rights.

We had planned a sensitivity analysis restricting the 2020 wave to participants who were personally affected (i.e., described themselves as worried and/or had taken at least two actions in response to the outbreak), but all 2020 participants met this criterion (see Results section below). An additional planned sensitivity analysis re-included all excluded data.

## Method

### Ethics statement

The Hillblom Aging Network protocol was reviewed and approved by the UCSF Committee on Human Research. This study was conducted in full compliance with the ethical principles set forth by the Declaration of Helsinki. All participants provided written informed consent.

### Participants

Our sample was drawn from community-dwelling older adults enrolled in the Hillblom Aging Network, a longitudinal study of healthy brain aging at the University of California, San Francisco (UCSF), Memory and Aging Center. Participants in this cohort are verified as neurologically normal based on a multidisciplinary assessment including a neurological examination, in-person neuropsychological testing, and an informant interview. As part of the Hillblom Aging Network, participants complete online web-based tasks in addition to in-person neuropsychological testing and neuroimaging.

Participants were recruited in two waves of data collection, prior to (2014 wave) and during the COVID-19 pandemic (2020 wave). Participants for the 2020 wave were recruited from the same longitudinal cohort of neurologically healthy older adults, restricted to cohort members that did not participate in the 2014 wave; thus, including participants either newly recruited to the broader cohort study since 2014, or who did not participate in the original data collection.

Some differences between the 2014 and 2020 waves include aesthetic differences in questionnaire formatting and preliminary instructions. Participants in the 2014 wave received a $20 Amazon gift card for completing the instrument; we did not have IRB approval for participant payment in the 2020 wave in the timeframe required for this study, so compensation was omitted. Members of both waves in this cohort received an email inviting them to participate in an online instrument that included the moral reasoning task described below.

## Materials

**Moral reasoning task.** In this task (S1 Appendix), participants read and were asked to make hypothetical decisions regarding twenty-four moral dilemmas from three categories as adapted by Chiong and colleagues [10], as part of a broader instrument on moral reasoning.

1. The Personal Rights (PR) category, composed of eight items, concerned choices in which the best overall outcome could only be produced by violating another individual's personal rights. For instance, whether to give a deadly injection to one person, which would identify a cure for an epidemic and save millions of people.

2. The Agent-centered Permissions (AP) category, composed of eight items, concerned choices in which the best overall outcome could be produced only at cost to the agent's own interests. For example, whether to make a donation or keep the money for one's own personal use.

3. The Special Obligation (SO) category, composed of eight items, concerned choices in which the best overall outcome could only be produced by forgoing opportunities to benefit the agent's family members, friends, or close others. For example, parental choices in which common-sense morality would prioritize one's own child's well-being over the well-being of other children.

They were then asked to rate, on a scale of 1–5, how confident they were in their decision and how emotional they were when making the decision. To ensure task engagement and comprehension, after each hypothetical decision participants were presented with a control question testing comprehension of the details of the dilemma situation (binary format yes/no).

**Questions about the personal impact of COVID-19.** We also collected responses to questions [24] about the impact of the COVID-19 outbreak on participants who responded in the 2020 wave. The first question assessed how worried respondents were about becoming sick, or about a friend or relative becoming sick, from the coronavirus. The second question assessed whether the participant had taken any of the following actions in response to the pandemic: a) decided not to travel or changed travel plans, b) bought or worn a protective mask, c) stocked up on items such as food, household supplies, or prescription medications, d) stayed home instead of going to work, school, or other regular activities, e) canceled plans to attend large gatherings, such as concerts or sporting events. These questions were used to confirm that participants in the 2020 wave had been personally affected by the outbreak.

## Procedure

The survey was administered online through Qualtrics. Participants first read information about the study, where prospective participants were advised about the sensitive nature of the task and that participation was optional. Each subsequent screen contained a moral dilemma, followed by questions assessing confidence/emotionality and a control question. Participants in the 2020 wave also received questions related to the COVID-19 pandemic following completion of the dilemmas. The last page provided a debriefing statement and thanked participants for taking the survey.

Utilizing a pre-registered analytic plan (https://osf.io/g2wtp), for Hypothesis 1 we summed each participant's utilitarian responses across the PR, AP, and SO categories. The total sum of utilitarian responses was entered as the outcome variable in a linear model with wave, age (mean centered), gender and educational attainment (mean centered) as predictors.

For Hypotheses 2 and 3, we extracted each participant's mean confidence and emotion elicitation scores across the PR, AP and SO moral categories. These scores were entered as outcome variables in linear models with wave, age, gender, and educational attainment as predictors. In pre-registered subsidiary analyses, we repeated these analyses using only

responses to the PR category of moral dilemmas. Analyses were performed using the statistical programming language R [25].

**Exclusion criteria.** We excluded data from participants that:

1. Incorrectly answered seven or more out of twenty-four control questions.

2. Did not give an answer to one (or more) moral dilemma.

3. Did not give an answer to one (or more) moral dilemma's follow up questions regarding emotionality and confidence.

## Results

We recorded responses of 281 older adults. We excluded 17 participants that incorrectly answered seven or more out of twenty-four control questions (2014 wave = 12, 2020 wave = 5) and 23 participants that did not give an answer to one (or more) moral dilemma (2014 wave = 10, 2020 wave = 13). For Hypotheses 2 and 3, we additionally excluded 55 participants that did not give an answer to one (or more) moral dilemma's follow up questions regarding emotionality (2014 wave = 11, 2020 wave = 23) and confidence (2014 wave = 10, 2020 wave = 11).

Some participants met more than one exclusion criterion and thus can be counted more than once (e.g., for not giving a response to one or more moral dilemma and for missing seven out of twenty-four control questions).

### Demographics

Descriptive statistics for the eligible sample are provided in Table 1. Reflecting the research cohort from which the sample was derived, the study sample was highly educated and predominantly white, with a mean age of 76.

### Personal impact of COVID-19

The majority of respondents in the 2020 wave were personally affected by the COVID-19 outbreak, most reporting worry and all adopting precautions against the pandemic (Tables 2 and 3; S1 Table).

**Table 1. Descriptive statistics: Age, education, gender, and race.**

| Demographics | 2014 wave | 2020 wave | Total | *p* |
|---|---|---|---|---|
| **Age** | | | | .243 |
| Mean (SD) | 76.8 (6.0) | 75.9 (6.1) | 76.3 (6.0) | |
| **Education** | | | | .329 |
| Mean (SD) | 17.8 (2.1) | 17.6 (1.9) | 17.7 (2.0) | |
| **Gender** | | | | .170 |
| Male | 66 (52.8%) | 58 (43.6%) | 124 (48.1%) | |
| Female | 59 (47.2%) | 75 (56.4%) | 134 (51.9%) | |
| **Race** | | | | .005 |
| White | 121 (96.8%) | 114 (85.7%) | 235 (91.1%) | |
| Asian | 4 (3.2%) | 11 (8.3%) | 15 (5.8%) | |
| Other Race | 0 (0.0%) | 4 (3.0%) | 4 (1.6%) | |
| Black/African American | 0 (0.0%) | 4 (3.0%) | 4 (1.6%) | |

*Note*: Descriptive statistics for participants who met inclusion criteria (*N* = 258) in the 2014 wave (*N* = 125) and 2020 wave (*N* = 133). Continuous variables of age and education are represented as mean (standard deviation), with *p* values from t-tests between waves; categorical variables of gender and race as count (percentages) with *p* derived from Fisher's exact test between waves. One observation was missing for the education variable (*N* = 257).

**Table 2. Descriptive statistics: Worry about COVID-19.**

| Worry | 2020 wave |
|---|---|
| Very worried | 22 (16.5%) |
| Somewhat worried | 73 (54.9%) |
| Not too worried | 33 (24.8%) |
| Not at all worried | 3 (2.3%) |
| (Missing) | 2 (1.5%) |

*Note*: Descriptive statistics for worry are represented as mean (percentages) for participants who met the inclusion criteria in the 2020 wave ($N = 133$). Participants were asked to respond the question: *How worried, if at all, are you that you or someone in your family or close friends will get sick from the coronavirus?*.

## Utilitarian judgments—overall and in personal rights dilemmas

In a linear regression analysis including wave, age, gender and educational attainment as predictors, utilitarian responses to dilemmas from all three categories (PR, AP and SO) were not significantly associated with wave (Table 4 and Fig 1).

However, utilitarian responses to dilemmas in the personal rights category differed significantly ($p = .001$) across waves (Table 5 and Fig 2); out of eight dilemmas in this category, participants in the 2020 wave gave 0.72 fewer utilitarian responses after adjusting for age, gender and education.

In sensitivity analyses re-including data from 17 excluded participants, who incorrectly answered seven or more out of twenty-four control questions, estimated coefficients and $p$ values differed only trivially (statistical significance of effects was unchanged). In addition, to examine whether the findings were driven by self-reported worry about COVID-19 pandemic, we repeated our analyses by splitting the sample (2020 wave) in two categories: Worried (by collapsing Very worried and Somewhat worried levels, see Table 2) and Not worried (by collapsing Not too worried and Not at all worried levels, see Table 2). Estimated coefficients and $p$ values differed only trivially with no significant effect of level of worry on utilitarian responses (overall and in personal rights dilemmas).

## Confidence and emotion elicitation—overall and in personal rights dilemmas

In responses to dilemmas from all three categories (PR, AP and SO), confidence and emotion elicitation did not differ significantly across waves ($p = .39$ and $p = .61$, respectively). Confidence was negatively associated with female gender ($b = -0.23$, $t(237) = -3.03$, $p = .003$) and emotion elicitation was positively associated with female gender ($b = 0.33$, $t(237) = 3.37$, $p <$

**Table 3. Descriptive statistics: Frequency of actions adopted against COVID-19.**

| N | Frequency | % Total | % Total Cum. |
|---|---|---|---|
| 5 | 70 | 52.63 | 52.63 |
| 4 | 45 | 33.83 | 86.47 |
| 3 | 13 | 9.77 | 96.24 |
| 2 | 5 | 3.76 | 100.00 |

*Note*: Descriptive statistics for the frequency of actions adopted are represented as count, frequency, percentages, and cumulative percent for participants who met the inclusion criteria in the 2020 wave ($N = 133$).

**Table 4. Utilitarian responses—All dilemmas.**

| Term | Estimate | SE | 95% CI | | t | p |
|---|---|---|---|---|---|---|
| | | | LL | UL | | |
| **Intercept** | 15.41 | 0.34 | 14.74 | 16.08 | 45.35 | < .001 |
| **Wave (2020)** | -0.39 | 0.39 | -1.17 | 0.38 | -1.00 | .32 |
| **Age (years)** | 0.04 | 0.03 | -0.02 | 0.11 | 1.33 | .18 |
| **Education (years)** | -0.02 | 0.10 | -0.22 | 0.18 | -0.17 | .87 |
| **Female gender** | 0.22 | 0.40 | -0.57 | 1.01 | 0.55 | .58 |

*Note*: The coefficient *Estimate* contains the intercept in the first row and the slopes (beta) at the following rows. *SE* represents the standard error, *LL* and *UL* the lower and upper limits of the confidence interval, *t* the t test statistic and *p* the probability value (*N* = 257).

.001) across waves (S2 and S3 Tables). In addition, confidence was negatively associated with educational attainment across waves ($b$ = -0.04, $t(237)$ = -2.07, $p$ = .04).

In analyses restricted to the personal rights category of dilemmas (S4 and S5 Tables), emotion elicitation and confidence did not differ significantly across waves ($p$ = .82 and $p$ = .35, respectively), but were associated with gender (emotion elicitation: $b$ = 0.48, $t(237)$ = 3.77, $p$ < .001; confidence: $b$ = -0.47, $t(237)$ = -4.61, $p$ < .001).

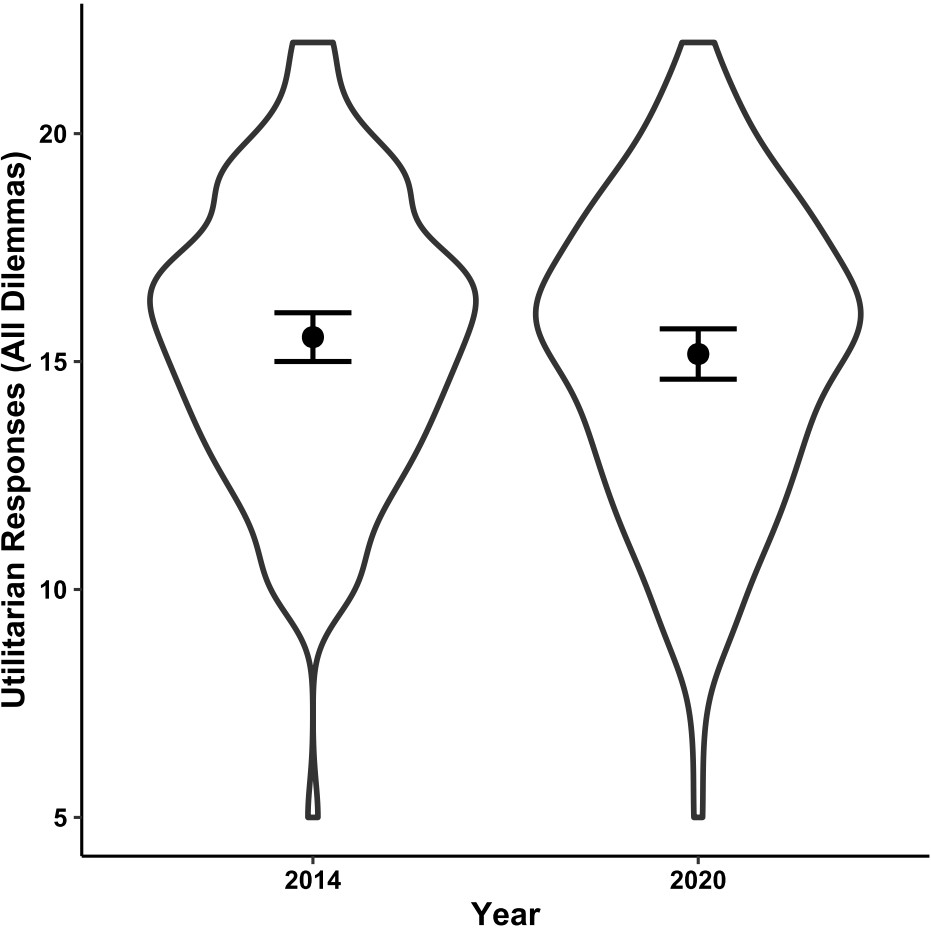

**Fig 1. Utilitarian responses—All dilemmas.** This violin plot represents the distribution of utilitarian responses in all three moral categories before (2014 wave) and during (2020 wave) the COVID-19 pandemic. The black dots represent the means with 95% confidence intervals (*N* = 258).

**Table 5. Utilitarian responses—Personal rights dilemmas.**

| Term | Estimate | SE | 95% CI | | t | p |
|---|---|---|---|---|---|---|
| | | | LL | UL | | |
| Intercept | 5.52 | 0.19 | 5.14 | 5.90 | 28.51 | < .001 |
| Wave (2020) | -0.72 | 0.22 | -1.16 | -.28 | -3.24 | .001 |
| Age (years) | 0.03 | 0.02 | 0.00 | 0.07 | 1.75 | .08 |
| Education (years) | 0.01 | 0.06 | -0.11 | 0.12 | 0.12 | .91 |
| Female gender | -0.08 | 0.23 | -0.53 | 0.37 | -0.35 | .73 |

*Note*: The coefficient *Estimate* contains the intercept in the first row and the slopes (beta) at the following rows. *SE* represents the standard error, *LL* and *UL* the lower and upper limits of the confidence interval, *t* the t test statistic and *p* the probability value (*N* = 257).

## Discussion

In this preregistered study, we examined individual judgments about hypothetical moral dilemmas during a global health crisis in a cohort of older adults at increased personal risk of

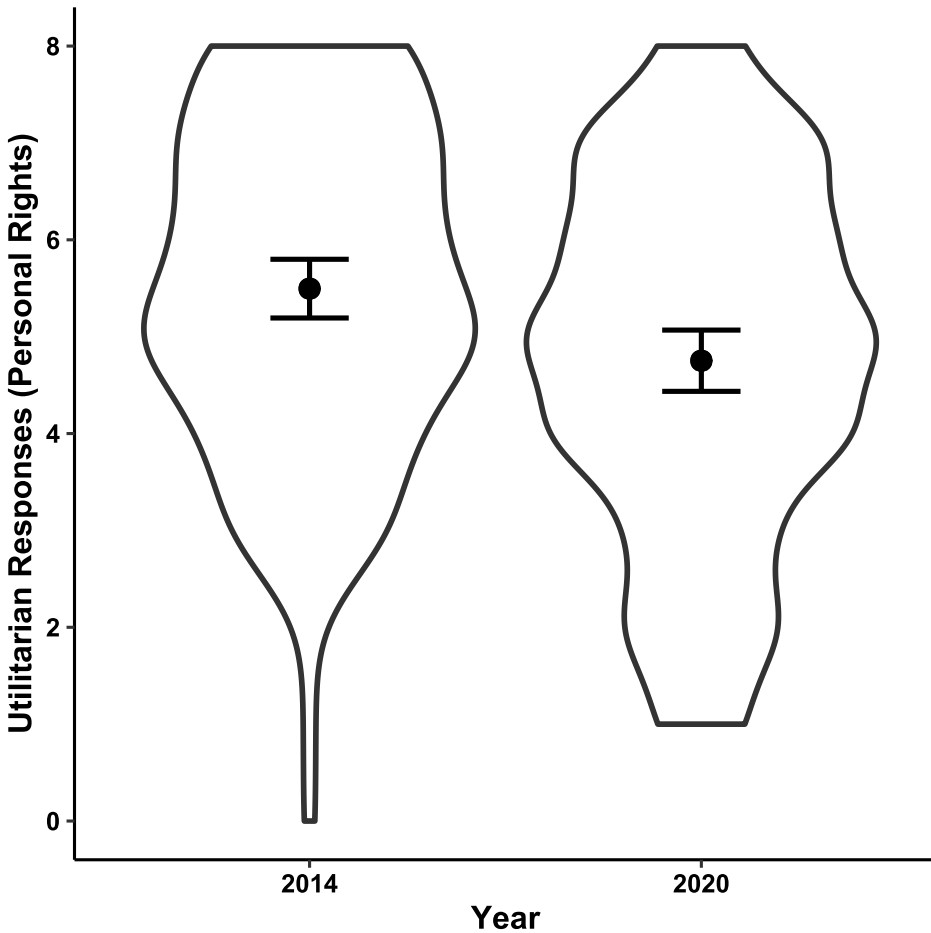

**Fig 2. Utilitarian responses—Personal rights dilemmas.** This violin plot represents the distribution of utilitarian decisions in the personal rights (PR) category of moral dilemmas before (2014 wave) and during (2020 wave) the COVID-19 pandemic. The black dots represent the means with 95% confidence intervals (N = 258).

severe complications from infection. We found that utilitarian judgments were not broadly affected by the COVID-19 pandemic, but that participants did make fewer utilitarian judgments about dilemmas involving conflicts between personal rights and the best overall outcome. Because emotion elicitation was not affected by the COVID-19 pandemic (both in analyses for all dilemmas and personal rights dilemmas), we did not find evidence that decreased utilitarian judgment in personal rights dilemmas was mediated by greater emotional elicitation in moral judgment, as suggested by dual-process theories. However, a dual-process specification of terror management theory posits that mortality salience and threat perception may promote non-utilitarian reasoning by depleting cognitive resources rather than by enhancing emotional elicitation, and our findings are congruent with the experimental manipulation by Trémolière and colleagues [6].

On the other hand, it is noteworthy that this finding was not observed for utilitarian judgments generally, but instead only for judgments about personal rights dilemmas. On dual-process accounts of moral cognition, it seems that utilitarian responses to agent-centered permissions (AP) and special obligations (SO) dilemmas should demand cognitive resources in a way analogous to personal rights (PR) dilemmas, and so should be similarly affected. This observed dissociation provides indirect support for psychological accounts that distinguish utilitarian willingness to harm from utilitarian impartial concern [12, 14].

Expressed confidence in moral judgments was also not affected by the COVID-19 pandemic, as expected by terror management theories. Gender was not a focus of our preregistered analysis, but we did observe that females reported less confidence in judgments and greater emotion elicitation than males; however, there was no effect of gender on utilitarian judgment whether examined across all dilemmas or only within personal rights dilemmas. Whether these differences reflect gender differences in the actual experience of confidence and emotion, or differences (e.g., due to socialization) in self-reporting of confidence and emotion, is unclear; particularly in the absence of gender differences in the primary outcome of interest (i.e., utilitarian decisions). Educational attainment was nominally associated with decreased reported confidence in moral judgments, although this effect was not a focus of our preregistered analysis, would not persist if adjusted for multiple comparisons, and also was not associated with differences in utilitarian judgment.

While age-related alterations in decision making are mostly associated with declines in deliberative abilities [26], whether moral judgements during the pandemic are susceptible to this effect is controversial. For instance, it has been hypothesized that older adults exhibit a reduced tendency to make utilitarian judgments due to working memory decline and affective processing improvement [27, 28]. Those accounts draw from dual process models of moral cognition and suggest that older adults make more deontological (i.e., less utilitarian) judgments due to the mediation of negative affective reactions [28]. Our study in both waves of data collection did not find a link between emotional elicitation and diminished utilitarian judgment (overall and in personal rights dilemmas), suggesting that age related differences in dual process frameworks do not explain our finding. Of note, in a recent study measuring demographic factors that may affect moral judgments during the pandemic, age was negatively associated with utilitarian judgments reflecting equitable public health and positively related with judgments maximizing human life expectancy [29], indicating possibly more complex influences of age on moral considerations in the current pandemic.

A principal limitation of the present study is the asymmetry of our samples between the 2014 and 2020 wave. For instance, most participants in the 2020 wave had changed their lives in at least two ways in response to the pandemic (increased level of worry and adopting a measure against COVID-19), which prevents having a true control group (i.e., older adults with no fear of the pandemic). Another important limitation constitutes the demographic

homogeneity of our study sample, which was largely white and highly educated. Racial and ethnic minorities comprise an estimated 23% of the older adult population within the United States [30], and Black and Latino populations have been disproportionately affected by the COVID-19 pandemic [31]. The generalizability of our findings to populations with the greatest burden of illness may then be limited, particularly given other known sociocultural influences on moral judgment [32]. Still, questions about the personal impact of the pandemic in the 2020 wave indicate that participants in our sample were affected, with most reporting worry and all reporting practical measures such as mask-wearing and refraining from regular activities outside the home.

A further limitation was that, because the 2020 wave was collected in an actual pandemic and not in the setting of a laboratory manipulation, some forms of experimental control were not available to exclude alternative explanations for differences between waves. The effects we observed could be attributable to other psychological, cultural, economic and political changes between task administration in 2014 and 2020. Also, given logistical constraints in promptly responding to the 2020 pandemic, we could not match all test features between waves (as in the gift card given to participants in the 2014 wave). This could produce biases due to differential study enrollment in the 2014 versus 2020 wave of data collection and altered responses to the moral dilemmas tested, where the direction of such effects are unknown. We note that the observed effect was large (adjusted 0.72 out of 8 possible responses); it would be unexpected and itself interesting if this effect could be attributed to more general trends over a short cultural timescale or subtle differences in task features. Meanwhile, our questions about personal impact confirm that the COVID-19 pandemic was highly significant to our participants, all of whom were older adults at high risk of death and disability from the coronavirus.

Our findings are in opposition with those reported by Francis and McNabb in a recent preprint [33]. These authors fielded a similar online instrument on a crowd working platform that included 14 personal rights dilemmas in August 2019 and in April 2020 yet did not find significant differences in the number of utilitarian responses prior to and during the COVID-19 pandemic. Several differences between our study and this study may account for this discrepancy in findings. The Francis and McNabb study utilized a within-subjects rather than between-subjects design in a smaller number of participants ($N = 83$). Because consistency in moral judgment is itself morally valued [34], administering the same questionnaire twice within 9 months could induce consistency effects. Participants in their study were substantially younger ($M = 35.5$, $SD = 12.6$) and therefore generally at much lower risk of death and major morbidity from the COVID-19 pandemic than participants in the present study.

It is important to contextualize our main finding, i.e., participants during the COVID-19 pandemic gave fewer utilitarian judgments in personal rights dilemmas. Meanwhile, another observed moral trend was broad adoption of utilitarian principles in facets of the healthcare domain [35], suggesting a shift from a more individualistic, notably deontological (non-utilitarian) medical approach to a focus on the net health benefit to populations (i.e., overall welfare). Our finding and this trend might not be mutually exclusive. Operating during a crisis with limited resources might demand clinicians to endorse utilitarian decisions (e.g., favoring treatment for younger over older individuals), without necessarily indicating an associated utilitarian willingness to violate personal rights. Non-utilitarian and utilitarian considerations may engage distinct social and individual ramifications depending on actors' roles in a health-crisis context. Other contextual factors, including regional variations in COVID-19 severity, could also affect the salience of different moral and public health considerations during this health crisis [29].

In summary, the present study represents a unique opportunity to examine utilitarian judgments during a global health crisis. Respondents during the COVID-19 pandemic gave fewer utilitarian responses to hypothetical dilemmas concerning conflicts between individual

personal rights and the best overall outcome. This finding indicates a reluctance to promote the best overall outcome when individual personal rights are at stake. This finding has important implications for policy decisions and public communication about pandemic-related issues such as clinical research, compulsory vaccination and enforced quarantines, in which such tradeoffs can arise. This finding is also noteworthy for psychological accounts of human moral reasoning that generate testable predictions about how global crises may affect individuals' judgments about cases. To that end, a reduced utilitarian willingness to violate individual rights for the greater good might reflect an intrinsic humanness and consideration for others as discrete individuals which is activated during times of crisis.

## Supporting information

**S1 Appendix. Moral reasoning task.** Moral reasoning task with the follow up questions on confidence/emotionality and the control question, categorized by moral category: Non-Moral (NM), Impersonal Moral (IM), Personal Right (PR), Agent-centered Permissions (AP), and Special Obligations (SO). Per our pre-registered analysis, only responses to PR, AP and SO dilemmas are reported in this paper.
(RTF)

**S1 Table. Descriptive statistics: Actions adopted against COVID-19.** Descriptive statistics for the actions adopted are represented as count (percentages) for participants who met the inclusion criteria in 2020 wave ($N = 133$).
(DOCX)

**S2 Table. Self-reported confidence—All dilemmas.** The coefficient *Estimate* contains the intercept in the first row and the slopes (beta) at the following ones. *SE* represents the standard error, *LL* and *UL* the lower and upper limits of the confidence interval, *t* the t test statistic and *p* the probability value ($N = 242$).
(DOCX)

**S3 Table. Self-reported emotionality—All dilemmas.** The coefficient *Estimate* contains the intercept in the first row and the slopes (beta) at the following ones. *SE* represents the standard error, *LL* and *UL* the lower and upper limits of the confidence interval, *t* the t test statistic and *p* the probability value ($N = 242$).
(DOCX)

**S4 Table. Self-reported emotionality—Personal rights dilemmas.** The coefficient *Estimate* contains the intercept in the first row and the slopes (beta) at the following ones. *SE* represents the standard error, *LL* and *UL* the lower and upper limits of the confidence interval, *t* the t test statistic and *p* the probability value ($N = 242$).
(DOCX)

**S5 Table. Self-reported confidence—Personal rights dilemmas.** The coefficient Estimate contains the intercept in the first row and the slopes (beta) at the following ones. SE represents the standard error, LL and UL the lower and upper limits of the confidence interval, t the t test statistic and p the probability value ($N = 242$).
(DOCX)

## Acknowledgments

The authors thank the Hillblom Aging Network study volunteers for their generous contributions to our research, particularly during the COVID-19 pandemic. We thank Agnieszka

Jaworska and Ryan Preston-Roedder for philosophical review of the content of tested dilemmas.

## Author Contributions

**Conceptualization:** Rea Antoniou, Heather Romero-Kornblum, Winston Chiong.

**Data curation:** J. Clayton Young, Michelle You.

**Formal analysis:** Rea Antoniou, Heather Romero-Kornblum.

**Methodology:** Heather Romero-Kornblum.

**Project administration:** Joel H. Kramer, Winston Chiong.

**Resources:** Joel H. Kramer, Winston Chiong.

**Supervision:** Winston Chiong.

**Visualization:** J. Clayton Young.

**Writing – original draft:** Rea Antoniou.

**Writing – review & editing:** Rea Antoniou, Heather Romero-Kornblum, J. Clayton Young, Michelle You, Joel H. Kramer, Winston Chiong.

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
