## [Decision Letter · Decision Letter 0]

25 Jun 2021

PONE-D-21-14926

No utilitarians in a pandemic? Shifts in moral reasoning during the COVID-19 global health crisis

PLOS ONE

Dear Dr. Antoniou,

Thank you for submitting your manuscript to PLOS ONE. After careful consideration, we feel that it has merit but does not fully meet PLOS ONE’s publication criteria as it currently stands. Therefore, we invite you to submit a revised version of the manuscript that addresses the points raised during the review process.

Please find below the reviewers' comments, as well as those of mine.

We look forward to receiving your revised manuscript.

Kind regards,

Valerio Capraro

Academic Editor

PLOS ONE

Journal Requirements:

Additional Editor Comments (if provided):

I have now collected two reviews from two experts in the field, whom I thank for their detailed and thoughtful reviews. Both reviewers think that the paper has potential, but they suggest major changes before it can be published. The main change regards the acknowledgement of the fact that you do not really have a control group, so you cannot make any causal statement. One of the reviewer even suggests to collect additional data. I don't feel strong about this, but I agree with the reviewer that the paper would greatly improve with more data. However, I won't make my decision conditional on this. Needless to say that also the other comments should be addressed. I also have a couple more comments, one regarding the section on dual process theories of moral reasoning. I have a review article on the topic, which you might find useful, as it covers virtually all the literature on the topic (Capraro, 2019). Also, the perspective article on what social and behavioural science can do to support pandemic response, published by Van Bavel et al in Nature Human Behaviour might be a useful general reference. Of course, it is not a requirement to cite these works, but I'm mentioning them because they look very related.

I am looking forward for the revision.

Van Bavel, J. J., et al. (2020). Using social and behavioural science to support COVID-19 pandemic response. Nature Human Behaviour, 4, 460-471.

Capraro, V. (2019). The dual-process approach to human sociality: A review. Available at SSRN 3409146.

Reviewers' comments:

Reviewer's Responses to Questions

**Comments to the Author**

1. Is the manuscript technically sound, and do the data support the conclusions?

Reviewer #1: Yes

Reviewer #2: Partly

2. Has the statistical analysis been performed appropriately and rigorously? 

Reviewer #1: Yes

Reviewer #2: Yes

3. Have the authors made all data underlying the findings in their manuscript fully available?

Reviewer #1: Yes

Reviewer #2: Yes

4. Is the manuscript presented in an intelligible fashion and written in standard English?

Reviewer #1: Yes

Reviewer #2: Yes

5. Review Comments to the Author

Reviewer #1: Review PONE-D-21-14926

The authors present a pre-registered study comparing a 2014 sample with a 2020 sample on moral utilitarianism. The authors rely on the terror management theory to predict that old people more at risk during the pandemic (2020 sample) should be less utilitarian more confident in their choices and should experience greater emotion than old people (associated in age) in 2014. The authors test these predictions on three categories of moral dilemmas (personal rights, agent-centered permissions, and special obligations). The main difference observed (which was also preregistered) was a difference to personal rights moral dilemmas (which commonly require to harm/kill someone for the greater good), a result consistent with the predictions from the dual-process specification of terror management theory. No differences between the two waves are observed for confidence and emotion elicitation.

The paper is well-written, concise, and the research question is clearly defined. Overall, I’m quite positive toward seeing this paper published, given some amendments, some of which are important however. Yet, I feel like it is a timely publication, which has the merit to crosspollinate the use of fictitious sacrificial dilemmas which largely lack ecological validity, yet with a real-world impacting context (COVID-19 pandemic). I have a few suggestions which I think should be addressed to increase the appeal of the paper.

An important weakness of the research is the absence of a true control group (old people with no fear of the COVID-19). The participants were drawn from the part of the Hillblom Aging Network, and I’m left puzzled why the authors did not try to collect additional data selecting participants from the network who did not exhibit anxiety signs regarding COVID-19. As a result, this decreases the appeal of the results. I leave the editor with the decision to collect additional data or not, but at least this should be included as a critical limitation (what if old people without fear/anxiety toward COVID-19 showed the same decrease in moral utilitarianism to sacrificial moral dilemmas?).

Another weakness of the paper is the discussion section. Regarding the context, much more could be said about moral utilitarianism during the COVID-19 pandemic. The results presented in this paper are interesting: old people are less utilitarian in the context of COVID-10 pandemic. Yet this is something quite opposed to what happens worldwide: health care provided have come in several states to endorse utilitarianism because of a lack of resources (for instance, by favoring the young at the expense of the older). Although I’m aware that this is not the very same topic, some discussion might be interesting and might help improve the discussion section. Finally, what is the take-home message of your research? Why is it an important one? (I personally think that it is an important research). The discussion section should answer more thoroughly such questionings.

References:

Vearrier, L., & Henderson, C. M. (2021, June). Utilitarian principlism as a framework for crisis healthcare ethics. In Hec Forum (Vol. 33, No. 1, pp. 45-60). Springer Netherlands.

Minor:

Unless this is a requirement from the journal, please replace “Material and method” by Method. Material is a subsection of Method.

Please format all the Tables according to the APA7 guidelines

I wish the authors good luck with their research.

Reviewer #2: This paper, “No utilitarians in a pandemic? Shifts in moral reasoning during the COVID-19 global health crisis” by Antoniou et al. presents the results of a pre-registered study analyzing moral judgements made by older adults in response to hypothetical dilemmas collected in two waves: one in 2014 and the other one during 2020. Dilemmas had three moral categories: Personal Rights (PRs), Agent-centered Permissions (APs), and Special Obligations (SOs).

Results show that participants’ responses did not change substantially when considering all three categories, but there was a significant difference in the direction hypothesized by the authors when restricting the analysis to just the PR category. Ratings of the emotion elicited by the decision did not significantly change.

There are several things to like about this paper, including the comparison of moral decisions in a time window pre- versus post- pandemic onset. The topic is timely and relevant and the statistical analyses are sound. The authors should also be commended for clearly outlining their hypotheses and pre-registering their analyses before collecting the second wave of data. However, there are several weak points in the manuscript which the authors need to address before I could recommend publication. These issues, some of which are major points, are entirely about the way in which the authors framed the paper and the interpretation of the observed results.

Major issues:

1- Causal interpretation: Several parts of the paper suggest that the authors are interpreting these findings in a causal way. Abstract: “We investigated the influence of…” (line 30) or Discussion: “we examined the effect of a global health crisis on…” (line 327). This study is 100% observational and the authors need to be clear and upfront about this. Moreover, they should avoid using terms such as “influence” or “effect” of the pandemic on moral choices given that the pandemic is not an experimental treatment. In one paragraph in the Discussion (lines 369-381), the authors acknowledge that many things happened between 2014 and 2020 and that the effect cannot be attributed to only the pandemic. (For example, could this study have been re-written as “No utilitarians during the Trump administration”?). Even if the results would have shown a clear and strong change in moral responses (which is not the case), this study cannot speak about the causal effect of the pandemic on moral choices (after all, it would be impossible to have such experimental treatment and the observational data presented here is clearly insufficient to suggest a causal role played by the pandemic). Similarly, this implies that this study cannot be thought of as a “real-world replication of the experimental manipulation in Tremoliere and colleagues” (line 339) since there is no experimental manipulation here.

2- Pre-registration: the main result inspiring the title and the take-home message of the paper is based on one of the three categories of moral dilemmas (PRs). The authors claim to have planned and pre-registered “subsidiary analyses focusing exclusively on the subset of dilemmas involving personal rights” (line 159). However, the registry in the Open Science Framework (https://osf.io/g2wtp) does not support that claim. There is no mention to such subsidiary analysis which suggests that the focus on dilemmas involving PRs was indeed exploratory. Therefore, the significant result found using PR dilemmas should be labeled in that way. There is nothing inherently wrong with the data not supporting the main hypothesis and to try publishing the results of a subsequent exploratory analysis, but the authors should be explicit about it. Given that this is a pre-registered study, every analysis that was not described in the registry should be labeled as exploratory.

3- Issues with the title: The title is misleading and problematic in at least three ways. First, the question “No utilitarians in a pandemic?” cannot be answered unless making a massive overstatement of the findings in the paper. The main result is that there is no overall change in utilitarian decision-making during the pandemic (i.e., results do not support the hypothesis H1 pre-registered by the authors), as 2 out of 3 categories of moral dilemmas show no significant difference.

Second, a shift in utilitarian decision-making does not imply that there is a shift in moral “reasoning”. As the authors correctly pointed out in lines 102-106, people might make utilitarian decisions for a variety of reasons that may or may not include engaging in utilitarian reasoning. In fact, DP theories (lines 107-126) suggest that making non-utilitarian decisions is consistent with a process different than reasoning, namely, emotional harm avoidance. So, one should not assume that the mechanism underlying the reduction in utilitarian choices is a “shift in moral reasoning” unless the data is accompanied by observations that support such mechanistic interpretation. Instead of presenting more data to test this mechanism, the authors could simply use mechanism-neutral language and refer to utilitarian “judgements” or “decisions”.

Third, it is also inaccurate to portrait participants making utilitarian decisions as “utilitarians”. While there are some stable individual differences in utilitarian thinking, recent studies in the literature suggest that terms such as “deontological” or “utilitarian” describe decisions made by people rather than people itself. This is because lay people (unlike philosophers) may make utilitarian choices without necessarily engaging in utilitarian thinking (Conway et al., 2018). Instead, it is more accurate to refer to participants as “people making utilitarian choices” or “utilitarian decision-makers” or “people making utilitarian judgements”, etc.

Other points:

4- External validity of the findings: This work studies moral responses made by individuals who are older than most participants in psychological research. This leaves open the question of whether and how these results are present in other populations. While the authors do not necessarily need to address this with more data, they should flag the possibility that age differences could play an important role in moral judgements. For example, a recent pre-print (Navajas et al., 2020, https://osf.io/ktv6z) found that moral decisions about dilemmas of the COVID-19 crisis correlate with utilitarian decision-making in hypothetical scenarios. Importantly, these utilitarian decisions became more prevalent in participants proceeding from countries where the crisis was more severe. Therefore, one could argue that such observations support the idea that a more dangerous context correlates with making more utilitarian decisions. This apparent discrepancy between empirical observations across studies could stem from the use of different methodologies but also from the measurement of utilitarian choices in populations which are very different in terms of age.

5- Limitations: When describing the limitations of the present work in the Discussion, the authors should also add the fact that there were different economic incentives to participate in the study in the 2014 versus 2020 wave of data collection.

6- Introduction (lines 52-60): please expand on the arguments made by competing theoretical predictions. What does it mean when Savulescu and colleagues claim “There are no egalitarian in a pandemic”?

6. PLOS authors have the option to publish the peer review history of their article (what does this mean?). If published, this will include your full peer review and any attached files.

Reviewer #1: No

Reviewer #2: No

---

## [Author Response · Author response to Decision Letter 0]

28 Jul 2021

Dear Dr. Valerio Capraro, 

Thank you for giving us the opportunity to submit a revised draft of the manuscript titled “No utilitarians in a pandemic? Shifts in moral reasoning during the COVID-19 global health crisis” to PLOS ONE. We appreciate the time and effort that you and the reviewers have dedicated to providing your valuable feedback on this manuscript. We are grateful to the reviewers for their insightful comments on the paper. We have been able to incorporate major changes to reflect the suggestions provided by the reviewers. Below you may find a point-by-point response to the reviewers’ comments and concerns.

Comments from Reviewer 1

Comment 1: An important weakness of the research is the absence of a true control group (old people with no fear of the COVID-19). The participants were drawn from the part of the Hillblom Aging Network, and I’m left puzzled why the authors did not try to collect additional data selecting participants from the network who did not exhibit anxiety signs regarding COVID-19.

Response 1: Thank you for this suggestion. In the preregistration (https://osf.io/g2wtp), for this reason we had proposed conducting a sensitivity analysis only including wave 2 participants who either (1) described themselves as either very worried or somewhat worried about the coronavirus outbreak, or (2) have taken at least two of the five described possible responses to the coronavirus outbreak. However, all respondents in wave 2 met criterion (2) of having taken at least two of the five possible responses to the pandemic, so this analysis would have been superfluous; more broadly, we conclude that all of the participants in wave 2 were practically affected by concerns about the pandemic. For this reason, the primary comparison was between the wave 1 cohort prior to the pandemic and the wave 2 cohort during the pandemic, although this does not exclude all other potential differences between cohorts. (At the same time, we note that a selected subgroup of older adults during COVID-19 that were unafraid of COVID-19 would itself have unique features distinguishing it from the wave 1 cohort, such that this might itself have introduced other uncontrolled differences in risk tolerance or anxiety.) To address the reviewer’s concerns given the patterns of response in our cohorts, we have conducted the following:

• While all of our wave 2 respondents met our prespecified criterion for being affected by the pandemic, we have conducted a further analysis to examine how different degrees of anxiety affect utilitarian responses (see “Result” section “Utilitarian judgments - overall and in personal rights dilemmas” lines 305-313). For this we split the sample of the 2020 wave in two categories: Worried (by collapsing the levels: Very worried and Somewhat worried) and Not worried (by collapsing the levels: Not too worried and Not at all worried). Estimated coefficients and p values differed only trivially with no significant effect of low level of worry on utilitarian responses.

• Emphasized this point and the asymmetry of our sample in our discussion (see “Discussion” section lines 368-372). Most participants in the 2020 wave had changed their lives in at least two ways in response to the pandemic (reporting worry and adopting a measure), which naturally evokes an asymmetry between the two samples.

Comment 2: Regarding the context, much more could be said about moral utilitarianism during the COVID-19 pandemic. The results presented in this paper are interesting: old people are less utilitarian in the context of COVID-10 pandemic. Yet this is something quite opposed to what happens worldwide: health care provided have come in several states to endorse utilitarianism because of a lack of resources (for instance, by favoring the young at the expense of the older). Although I’m aware that this is not the very same topic, some discussion might be interesting and might help improve the discussion section.

Response 2: We agree with this suggestion and we have, accordingly, revised the discussion to emphasize this point (see “Discussion” section lines 420-430). We added a paragraph combining our findings and the points raised in the proposed reference: “Vearrier, L., & Henderson, C. M. (2021, June). Utilitarian principlism as a framework for crisis healthcare ethics. In Hec Forum (Vol. 33, No. 1, pp. 45-60). Springer Netherlands.” 

Comment 3: Finally, what is the take-home message of your research? Why is it an important one? The discussion section should answer more thoroughly such questionings.

Response 3: We agree with this and we have included your suggestion throughout the in the discussion section. We elaborated the discussion by emphasizing why this research is important and which the associated implications are (see “Discussion” section lines 431-442).

Comment 4: Unless this is a requirement from the journal, please replace “Material and method” by Method. Material is a subsection of Method. 

Response 4: According to the journal’s requirements this element (Material and Method) can be renamed as needed. We have thus incorporated your suggestion throughout the manuscript (see “Method” section line 167).

Comment 5: Please format all the Tables according to the APA7 guidelines

Response 5: Thank you so much for your suggestion. We have followed the journal’s guidelines for tables (https://journals.plos.org/plosone/s/tables and https://journals.plos.org/plosone/s/file?id=wjVg/PLOSOne_formatting_sample_main_body.pdf), which depart slightly from the APA7 guidelines. (For example, the journal’s guidelines call for the title to be on the same line as the table number, and also display borders and lines separating each cell.)

Comments from Reviewer 2

Comment 1: Several parts of the paper suggest that the authors are interpreting these findings in a causal way. Abstract: “We investigated the influence of…” (line 30) or Discussion: “we examined the effect of a global health crisis on…” (line 327). This study is 100% observational and the authors need to be clear and upfront about this. Moreover, they should avoid using terms such as “influence” or “effect” of the pandemic on moral choices given that the pandemic is not an experimental treatment. In one paragraph in the Discussion (lines 369-381), the authors acknowledge that many things happened between 2014 and 2020 and that the effect cannot be attributed to only the pandemic. Even if the results would have shown a clear and strong change in moral responses (which is not the case), this study cannot speak about the causal effect of the pandemic on moral choices (after all, it would be impossible to have such experimental treatment and the observational data presented here is clearly insufficient to suggest a causal role played by the pandemic). Similarly, this implies that this study cannot be thought of as a “real-world replication of the experimental manipulation in Tremoliere and colleagues” (line 339) since there is no experimental manipulation here.

Response 1: Thank you for pointing this out. We agree with this comment. Therefore, we have changed the causal tone (by replacing words like effect or influence) with words that reflect better the observational nature of our study (e.g., context):

• in the “Abstract” (see lines 29-31): “We investigated moral judgments in the context of the contemporary global crisis among older adults, who are at greatest personal risk from the pandemic.”

• in the “Hypotheses and research questions” (see lines 143-144): “The current study utilizes a unique opportunity to study utilitarian judgments within the context of COVID-19 pandemic.”

• in the “Discussion” (see lines 336-338, 428-430, 431-434): e.g., “In this preregistered study, we examined individual judgments about hypothetical moral dilemmas during a global health crisis in a cohort of older adults at increased personal risk of severe complications from infection.” & “Non-utilitarian and utilitarian considerations may engage distinct social and individual ramifications depending on actors’ roles in a health-crisis context.” & “In summary, the present study represents a unique opportunity to examine utilitarian judgments during a global health crisis. Respondents during the COVID-19 pandemic gave fewer utilitarian responses to hypothetical dilemmas concerning conflicts between individual personal rights and the best overall outcome.”

We have accordingly removed our suggestion that this study can be thought of as a “real-world replication of the experimental manipulation in Tremoliere and colleagues” (see lines 347-348)

Comment 2: The main result inspiring the title and the take-home message of the paper is based on one of the three categories of moral dilemmas (PRs). The authors claim to have planned and pre-registered “subsidiary analyses focusing exclusively on the subset of dilemmas involving personal rights” (line 159). However, the registry in the Open Science Framework (https://osf.io/g2wtp) does not support that claim. There is no mention to such subsidiary analysis which suggests that the focus on dilemmas involving PRs was indeed exploratory. Therefore, the significant result found using PR dilemmas should be labeled in that way. There is nothing inherently wrong with the data not supporting the main hypothesis and to try publishing the results of a subsequent exploratory analysis, but the authors should be explicit about it.

Response 2: We appreciate this comment; the analysis restricted to PR dilemmas is labeled a “sensitivity analysis” in the preregistration (second to last paragraph): “we will repeat the primary analyses for hypotheses 1-3, but restricted to responses to PR dilemmas (which have been the main focus of dual process moral theories to date).” For this reason, we have first reported the primary analysis including dilemmas from all categories, before reporting the PR results. The PR-restricted analyses are planned and not post hoc. 

Comment 3: The title is misleading and problematic.

Response 3: We agree, and we changed the title accordingly to reflect better the take home message of our study: “Reduced Utilitarian Willingness to Violate Personal Rights during the COVID-19 Pandemic”. See lines 3-4. We additionally have changed the subtitle of the study to: “Utilitarian Judgements during COVID-19” (see heading throughout the manuscript).

Comment 4: As the authors correctly pointed out in lines 102-106, people might make utilitarian decisions for a variety of reasons that may or may not include engaging in utilitarian reasoning. In fact, DP theories (lines 107-126) suggest that making non-utilitarian decisions is consistent with a process different than reasoning, namely, emotional harm avoidance. So, one should not assume that the mechanism underlying the reduction in utilitarian choices is a “shift in moral reasoning” unless the data is accompanied by observations that support such mechanistic interpretation. Instead of presenting more data to test this mechanism, the authors could simply use mechanism-neutral language and refer to utilitarian “judgements” or “decisions”.

Response 4: Thank you for pointing this out. We have accordingly replaced the moral reasoning term to utilitarian judgments in:

• subheadings of the “Introduction” (see line 84)

• the section “Hypotheses and research questions” (see lines 143-144)

• in the “Discussion” (see lines 431-432)

Comment 5: This work studies moral responses made by individuals who are older than most participants in psychological research. This leaves open the question of whether and how these results are present in other populations. While the authors do not necessarily need to address this with more data, they should flag the possibility that age differences could play an important role in moral judgements.

Response 5: Thank you for pointing this out. We have incorporated an additional paragraph in the “Discussion” addressing this issue. See lines 382-391.

Comment 6: In the discussion authors should also add the fact that there were different economic incentives to participate in the study in the 2014 versus 2020 wave of data collection.

Response 6: Thank you for pointing this out. See in the “Discussion” section lines 396-401: “Also, given logistical constraints in promptly responding to the 2020 pandemic, we could not match all test features between waves (as in the gift card given to participants in the 2014 wave). This could produce biases due to differential study enrollment in the 2014 versus 2020 wave of data collection, though at first glance the absence of a personal financial incentive in the second wave might have been expected to select for a more rather than less utilitarian cohort.”

Comment 7: Please expand on the arguments made by competing theoretical predictions. What does it mean when Savulescu and colleagues claim “There are no egalitarian in a pandemic”? (lines 52-60)

Response 7: Thank you for this comment; we have elaborated on the relevance of Savalescu’s claim to our research question. See lines 55-58 “In a recent article advocating for utilitarian approaches to this crisis Savulescu and colleagues [4] have claimed that “There are no egalitarians in a pandemic,” suggesting the empirical psychological claim that health crises may reduce the salience of non-utilitarian moral considerations.”

Additional clarifications

In addition to the above comments, we appreciate your suggestion to collect more data. However, given the unique opportunity to test participants inside and outside of a historic global pandemic, we were not able to collect further data that illuminate our key questions. 

We look forward to hearing from you in due time regarding our submission and to respond to any further questions and comments you may have.

---

## [Decision Letter · Decision Letter 1]

6 Sep 2021

PONE-D-21-14926R1No utilitarians in a pandemic? Shifts in moral reasoning during the COVID-19 global health crisisPLOS ONE

Dear Dr. Antoniou,

Thank you for submitting your manuscript to PLOS ONE. After careful consideration, we feel that it has merit but does not fully meet PLOS ONE’s publication criteria as it currently stands. Therefore, we invite you to submit a revised version of the manuscript that addresses the points raised during the review process.

We look forward to receiving your revised manuscript.

Kind regards,

Valerio Capraro

Academic Editor

PLOS ONE

Additional Editor Comments:

One of the reviewers feel that their comments were not satisfactorily addressed and (re)suggest major revision. Therefore, I would like to invite you to revise your manuscript again. In your response letter, please respond to all reviewer's comments.

I am looking forward for the revision.

Reviewers' comments:

Reviewer's Responses to Questions

**Comments to the Author**

1. If the authors have adequately addressed your comments raised in a previous round of review and you feel that this manuscript is now acceptable for publication, you may indicate that here to bypass the “Comments to the Author” section, enter your conflict of interest statement in the “Confidential to Editor” section, and submit your "Accept" recommendation.

Reviewer #2: (No Response)

2. Is the manuscript technically sound, and do the data support the conclusions?

Reviewer #2: Partly

3. Has the statistical analysis been performed appropriately and rigorously? 

Reviewer #2: Yes

4. Have the authors made all data underlying the findings in their manuscript fully available?

Reviewer #2: Yes

5. Is the manuscript presented in an intelligible fashion and written in standard English?

Reviewer #2: Yes

6. Review Comments to the Author

Reviewer #2: The authors have addressed most of my previous concerns.

First, I would like to apologize for missing the part of the pre-registration where they stated the subsidiary analysis reported in this work. I do believe now that, although it was not the main point of the pre-registration, the result based on PR dilemmas was planned.

I have, however, two more comments about this revised version of the manuscript:

1- When you discuss the potential role of incentives in partially explaining the different results between cohorts you state: “though at first glance the absence of a personal financial incentive in the second wave might have been expected to select for a more rather than less utilitarian cohort.” Do we know how people vary in their responses to PR dilemmas based on whether or not they receive economic incentives due to their participation? While I understand the point the authors wanted to make, I do not see anything in the data or in the literature supporting this statement. Instead, I believe authors should simply acknowledge they do not know the magnitude or direction of the effect of financial incentives on their data.

2- The authors claim that “whether moral judgements are susceptible to this [age-related] effect remains unclear”. In my previous report, I pointed the authors to a recent study (Navajas et al., 2020, https://osf.io/ktv6z) that examines how demographic factors (including age) correlate with utilitarian judgements during the pandemic. [The part of my original report where I suggested authors to look at this paper was removed and did not appear in the response letter]. Discussing the relationship with those results would help better situating the findings reported in this work in the context of a fast-changing literature.

7. PLOS authors have the option to publish the peer review history of their article (what does this mean?). If published, this will include your full peer review and any attached files.

Reviewer #2: No

---

## [Author Response · Author response to Decision Letter 1]

1 Oct 2021

Dear Dr. Valerio Capraro, 

Thank you for giving us the opportunity to re-submit a revised draft of the manuscript titled “Reduced Utilitarian Willingness to Violate Personal Rights during the COVID-19 Pandemic” to PLOS ONE. We appreciate the time and effort that you and the reviewers have dedicated to providing your valuable feedback on this manuscript. We are grateful to the reviewers for their insightful comments on the paper. We have been able to incorporate the changes to reflect the suggestions provided by reviewer #2. Below you may find a point-by-point response to reviewers’ 2 comments and suggestions.

Comments from Reviewer #2

Comment 1: When you discuss the potential role of incentives in partially explaining the different results between cohorts you state: “though at first glance the absence of a personal financial incentive in the second wave might have been expected to select for a more rather than less utilitarian cohort.” Do we know how people vary in their responses to PR dilemmas based on whether or not they receive economic incentives due to their participation? While I understand the point the authors wanted to make, I do not see anything in the data or in the literature supporting this statement. Instead, I believe authors should simply acknowledge they do not know the magnitude or direction of the effect of financial incentives on their data.

Response 1: Thank you for this suggestion. We agree with this suggestion and we have, accordingly, revised the paragraph and removed this conjecture (see “Discussion” section lines 407-410): “This could produce biases due to differential study enrollment in the 2014 versus 2020 wave of data collection and altered responses to the moral dilemmas tested, where the direction of such effects are unknown. ”

Comment 2: The authors claim that “whether moral judgements are susceptible to this [age-related] effect remains unclear”. In my previous report, I pointed the authors to a recent study (Navajas et al., 2020, https://osf.io/ktv6z) that examines how demographic factors (including age) correlate with utilitarian judgements during the pandemic. [The part of my original report where I suggested authors to look at this paper was removed and did not appear in the response letter]. Discussing the relationship with those results would help better situating the findings reported in this work in the context of a fast-changing literature. 

Response 2: Thank you for this suggestion. We agree with this and we have referenced this study in the discussion section. 

• We elaborated the discussion by emphasizing how demographics may affect utilitarian judgements during the pandemic (see “Discussion” section lines 382-386): “Of note, in a recent study measuring demographic factors that may affect moral judgments during the pandemic, age was negatively associated with utilitarian judgments reflecting equitable public health and positively related with judgments maximizing human life expectancy [29], indicating possibly more complex influences of age on moral considerations in the current pandemic.” 

• Drawing from the Navajas 2020 paper, we additionally included a small section addressing other contextual factors such as COVID-19 severity in different countries (lines:438-440) “Other contextual factors, including regional variations in COVID-19 severity, could also affect the salience of different moral and public health considerations during this health crisis [29].

Additional clarifications

In addition to the above comments, we have updated the number cases in the introduction section due to COVID-19 pandemic (see line 43).

We look forward to hearing from you in due time regarding our re-submission and to respond to any further comments you may have.

---

## [Decision Letter · Decision Letter 2]

13 Oct 2021

Reduced Utilitarian Willingness to Violate Personal Rights during the COVID-19 Pandemic

PONE-D-21-14926R2

Dear Dr. Antoniou,

We’re pleased to inform you that your manuscript has been judged scientifically suitable for publication and will be formally accepted for publication once it meets all outstanding technical requirements.

Kind regards,

Valerio Capraro

Academic Editor

PLOS ONE

Additional Editor Comments (optional):

Reviewers' comments:

Reviewer's Responses to Questions

**Comments to the Author**

1. If the authors have adequately addressed your comments raised in a previous round of review and you feel that this manuscript is now acceptable for publication, you may indicate that here to bypass the “Comments to the Author” section, enter your conflict of interest statement in the “Confidential to Editor” section, and submit your "Accept" recommendation.

Reviewer #2: All comments have been addressed

2. Is the manuscript technically sound, and do the data support the conclusions?

Reviewer #2: Yes

3. Has the statistical analysis been performed appropriately and rigorously? 

Reviewer #2: Yes

4. Have the authors made all data underlying the findings in their manuscript fully available?

Reviewer #2: Yes

5. Is the manuscript presented in an intelligible fashion and written in standard English?

Reviewer #2: Yes

6. Review Comments to the Author

Reviewer #2: The authors have addressed all my remaining comments. Thank you.

7. PLOS authors have the option to publish the peer review history of their article (what does this mean?). If published, this will include your full peer review and any attached files.

Reviewer #2: No

---

## [Editor Report · Acceptance letter]

15 Oct 2021

PONE-D-21-14926R2 

Reduced Utilitarian Willingness to Violate Personal Rights during the COVID-19 Pandemic 

Dear Dr. Antoniou:

I'm pleased to inform you that your manuscript has been deemed suitable for publication in PLOS ONE. Congratulations! Your manuscript is now with our production department. 

Kind regards, 

on behalf of

Dr. Valerio Capraro 

Academic Editor

PLOS ONE